

Using climate change scenarios to simulate mobility of metal contaminants in soils: the example of copper on
a European scale
Laura SERENI[1,2], Julie-Maï PARIS[3], Isabelle LAMY[1], Bertrand GUENET[3]
[1]Université Paris-Saclay, INRAE, AgroParisTech, UMR EcoSys, 91120 Palaiseau, France
[2]Present address: Univ. Grenoble Alpes, CNRS, INRAE, IRD, Grenoble INP, IGE, Grenoble, France
[3]Laboratoire de Géologie ENS, PSL Research University, CNRS, UMR 8538, IPSL, Paris, France
*Correspondence to Laura Sereni (laurasereni@yahoo.fr)
Abstract:
Soil contaminant deposition is highly dependent on anthropogenic activities while contaminant retention,
mobility and availability are highly dependent on soil properties. The knowledge of partitioning between soil solid
and solution phases is necessary to estimate whether deposited amounts of contaminants will rather be leached
through runoff or accumulated. Besides pedological driven partitioning, runoff is expected to change during the
next century due to changes in climate and in rainfall patterns. In this study, we aimed at estimating at the
European scale the areas concerned by potential risk due to contaminant leaching (LP). We also defined in the
same way the surface areas where limited Cu leaching occurred, leading to potential accumulation (AP) areas.
Among contaminant, we focused on copper (Cu) widely used in agriculture, resulting in high spatial variations in
deposited and incorporated amounts in soils. We developed a method using both Cu partition coefficients ($K_f$)
between total and dissolved Cu forms, and runoff simulation results for historical and future climates. The
calculation of $K_f$ with pedo-transfer functions allowed us to avoid any uncertainties due to past management or
future depositions that may affect total Cu concentrations. Areas with high potential risk of leaching or of



accumulation were estimated over the XXI[th] century by comparing $K_f$ and runoff to their respective European
median. Thus, at three distinct times, we considered a grid point at risk of LP if its $K_f$ was low compared to the
European median and its runoff was high compared to the European median of the time. Similarly, a grid point
was considered at risk of AP if its $K_f$ was high and its runoff was low compared to their respective European median
of the time. To deal with uncertainties in climate change scenarios and the associated model projections, we
performed our study with two representative atmospheric greenhouse gases concentration pathways, defined
with climate change associated to a large set of socio-economic scenarios found in the literature. We used two
land surface models (ORCHIDEE and LPJmL, given soil hydrologic properties) and two global circulation models
(ESM2m and CM5a, given rainfall forecast). Our results show that, for historical scenario 6.4 ± 0.1 % (median,
median deviation) and 6.7 ± 1.1 % of the grid cells of the European land surfaces are concerned by LP and AP
respectively. Interestingly, our results simulate a constant global surface concerned by LP and AP, around 13% of
the grid cells, consistent with an increase in AP and a decrease in LP. Despite large variations in LP and AP extents
depending on the land surface model used for estimations, the two trends were more pronounced with RCP 6.0
than with RCP 2.6, highlighting the global risk of combined climate change and contamination and the need for
more local assessment. Results are discussed to highlight the points requiring improvement to refine predictions.
Keywords: regional modeling, transfer functions, ISIMIP, LUCAS Topsoil data, mapping risk



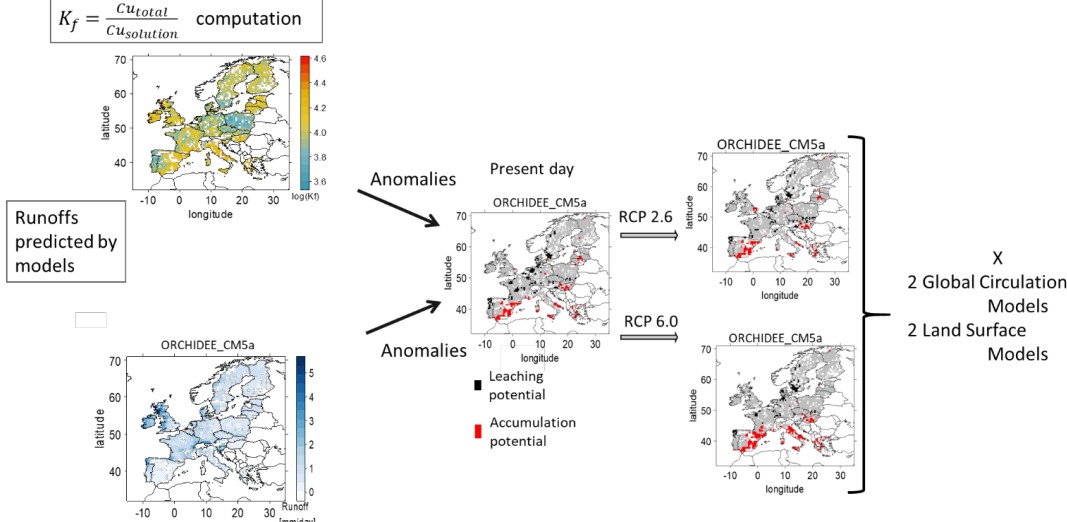






1. Introduction

At a large spatial scale, trace element contents in soils are highly variable in relation with the trace element contents of the soil parental rocks and with local anthropogenic inputs of various origins (Flemming and Trevors, 1989; Noll, 2003; Salminen and Gregorauskiene, 2000). Some trace elements like copper (Cu) or zinc are required for several biological mechanisms, but when highly concentrated they may have toxic effects on soil organisms (Giller et al., 1998). In particular, Cu is widely used, as a fungicide, especially against downy mildew in vineyards (Komárek et al., 2010), but also in industrial processes. At the European scale, a gradient of soil Cu concentrations can be find from typical baseline values between 5 mgCu.kg$^{-1}$ to 20 mgCu.kg$^{-1}$ (Salminen and Gregorauskiene, 2000), to values larger than 100 mgCu.kg$^{-1}$, common in cultivated soils and especially in vineyards parcels (Ballabio et al., 2018). It is commonly accepted to conceptually partition the total soil Cu content into different pools of Cu forms in close equilibrium. Briefly, three pools can be defined: a so-called 'inert' pool corresponding to Cu included into minerals, a so-called 'labile' pool corresponding to Cu sorbed to soil constituents but that can be mobilized according to environmental conditions, and a smallest 'mobile' pool corresponding to Cu in soil solution that may be readily available for living organisms but also for transport within soil horizons (Broos et al., 2007; Rooney et al., 2006; West and Coombs, 1981). Schematically, these pools are governed by processes like exchange, complexation or sorption. Also, local soil characteristics such as organic matter, pH or cationic exchange capacity can affect the proportion of Cu in these different pools (Vidal et al., 2009). Any modifications in soil properties or soil solution composition may thus affect Cu equilibrium between sorbed and solution phases. The pool of Cu in the solution phase can be assimilated to a potential pool of Cu leaching. Conversely, Cu bound to the solid phases can be assimilated to a potential pool of Cu accumulation in soil. Depending on the main process involved, for a given amount of Cu deposited on soil, the proportions of leached and accumulated Cu can varied from place to place and with time. However, studies simulating whether the soil will rather leach or accumulate a contaminant, i.e. will act as a source or a sink for the contamination, are scarce especially at a large spatial scale. This knowledge, however, could allow to highlight contaminated areas with a potential to leach, disperse or accumulate



contaminants, and therefore help for long term environmental management.
Concurrently, climate change due to anthropogenic activities is expected to impact rainfall patterns in the
forthcoming the decades, leading to changes in the frequency and intensity of weather events at regional and
local levels (Christensen and Christensen, 2003). For instance, projections forecast an increase in rain- and snow-
fall events in winter in Northern Europe but a decrease in summer in the Mediterranean region, which extends to
northward regions (Douville et al., 2021). The extent of rain- and snow-fall alterations depends on anthropogenic
activities and associated climate change. Thus,  climate change will alter water flows throughout the century
(Mimikou et al., 2000). For instance, increase in rainfall intensity and in water accumulation in the soil surface due
to limited water infiltration may induce large runoff (Chu et al., 2019). Changes in runoff will also change fluxes
of elements or of particulates in the soil solution as it has been shown for Cu (Babcsányi et al., 2016). However,
the relationships between these changes in runoff and fluxes of elements is still poorly predicted for the next
decades.
In this framework, our aim was twofold: i) estimate the soil Cu leaching potential areas in Europe, thereafter,
named LP, for the beginning of the century and ii) predict their evolution according to different climate change
scenarios. Additionally, we aimed to estimate the Cu accumulation potential areas thereafter named AP. We
hypothesized that the role of soil as Cu sink or source, linked to the processes of Cu accumulation or leaching, can
be described by the combined effects of local runoff amounts and of local soil properties controlling the partition
of total Cu in sorbed and solution species. Due to the lack of information about the future Cu deposition and on
soil Cu concentrations whatever its form, we developed a method using the partition coefficient ($K_f$) at the
equilibrium between solid and solution phases to determine areas with high or low potential of leaching whatever
total Cu concentration. Regarding the lack of data about future deposited amount at large scale, using $K_f$ was
necessary to estimate the Cu mobility potential. The LP or AP areas were thus estimated through the combined
use of $K_f$, calculated with the help of pedo-transfer functions, and the use of soil runoff amounts extracted from
earth system simulations. With the use of $K_f$ we avoided the uncertainties due to past land management and



previous Cu deposition and focused on risks arising from future deposition. To do so, we first reviewed the
empirical equations estimating Cu's $K_f$ based on soil properties to highlight generic soil properties governing this
partition. From this review, we extracted the best compromise $K_f$ equation to estimate partitioning at the regional
scale, which ensures more accurate Kf calculation based on pedo-geochemical data typically recorded in soil
surveys, thus mainly available. This allowed us to estimate Cu's $K_f$ values to be used at the European scale based
on pedo-geochemical soil surveys without the knowledge of soil Cu total content. We then focused on the current
state of the climate and its projected changes over the XXI$^{st}$ century, based on two climate change scenarios. To
capture the difficulties in runoff prediction and to disentangle the uncertainties between rainfall prediction and
runoff calculations of land surface models, we used a set of simulations provided by the Inter-Sectoral Impact
Model Intercomparison Project (ISIMIP). These simulations used different land surface models driven by different
climate forcings computed by different climate models.  For each scenario and each couple of land surface model
and climate forcing we estimated the LP or AP of each grid points by comparison between the local values of $K_f$
and of runoff to the respective calculated European median.

2.  Materials and methods

2.1. Equations to estimate copper $K_f$
The rigorous definition of $K_f$ is based on the concentration ratio of sorbed vs solution species (here Cu) at the
equilibrium. Yet, for practical reasons of measurement and applicability, $K_f$ is conventionally derived from total Cu
and not from sorbed Cu (Degryse et al., 2009). A general form of the Cu partition coefficient between soil and
solution – $K_f$ – can be used to describe Cu concentrations in the sorbed and solution phases, defined as Eq. (1):

$$K_f = \frac{Cu_{total}}{Cu_{solution}n} \quad (1)$$



Where *n* stands for the variation in binding strength with metal loading (Groenenberg et al., 2010). A low $K_f$
reflects a high proportion of Cu in solution for a given total Cu content of the soil. $K_f$ can vary as a function of
different soil parameters (Degryse et al., 2009; Elzinga et al., 1999) and can also be estimated using Eq. (2):
$log_{10}(K_f) = a_0 + \sum_i a_i log_{10}(X_i)$  (2)
with $X_i$ the different soil parameters and $a_i$ the corresponding associated coefficient to the parameter.
Numerous studies in the literature have attempted the determination of the value of $K_f$. using the Eq. (2) based
on statistical relationships between soil pedo-geochemical parameters, Cu in solution and total Cu measurements.
The soil pedo-geochemical parameter $X_i$ and its associated coefficient $a_i$ can differ depending on the study and
the data set used for the estimation.  For the purposes of this study, $K_f$ is estimated at the European level, so the
formula chosen strikes the best balance between the accuracy of the relationship and its applicability on a wide
scale. Thus, the equation must:

125         i)          Include only parameters that are measured in large soil surveys

126         ii)         Fit a large range of each soil parameter

127         iii)        Focus on in situ long-term contamination and not on laboratory experiments.


On December 2020 we first ran a bibliographic research on WOS looking for "Cu AND availab*AND soil AND TOPIC
function". We then completed this research by examining the references cited in the articles found. We collected
the available relationships for estimating $K_f$ on the basis of soil pedo-geochemical characteristics and/or total Cu.
We selected only relationships that were based on commonly collected soil pedo-geochemical characteristics, such
as soil organic matter (OM) or soil organic carbon (OC), dissolved organic carbon (DOC), cationic exchange capacity
(CEC), clay percentage and pH that are the most frequently reported values from large scale soil survey.

2.2 Soil data



This study used European data on various soil parameters, in particular pH and organic carbon (OC), obtained from
the Joint Research Centre's (JRC) LUCAS topsoil data. The data set is limited only to the territories of European Union
Member     States.     The     aforementioned     data     set     provides     information     on     pH
(https://esdac.jrc.ec.europa.eu/content/copper-distribution-topsoils)     and     OC     contents
(https://esdac.jrc.ec.europa.eu/content/topsoil-soil-organic-carbon-lucas-eu25).  The  data  has  been  re-gridded
with cdo commands (Schulzweida, 2019) to a spatial resolution of 0.5° (equivalent to approximately 50 km). This
was done to match the resolution of the land surface models that were used to estimate the runoff. The resulting
runoff data is presented in section 2.3.
2.3. Runoff data from land surface models
Runoff is computed in land models from incoming rain- and snow- falls, calculated evapotranspiration, and soil
hydrologic capacities. To estimate changes in runoff across century and to reduce uncertainties, we used two typical
land-surface schemes models (LSM) – namely ORCHIDEE (Krinner et al., 2005) and LPJmL (Sitch et al., 2003)– and
two global circulation models (GCM) providing climate projections – namely IPSL-CM5a (Dufresne et al., 2013) and
GDFL-ESM2m (Dunne et al., 2012) – further named CM5a and ESM2m respectively. Our study exploited simulations
conducted as part of the Inter-Sectoral Impact Model Intercomparison Project Phase 2b (ISIMIP2b), which supplied
simulations of land surface models driven by binding scenarios from 1861 to 2099 (Frieler et al., 2017). Further
details of the protocol used can be found at https://www.isimip.org/protocol/#isimip2b. The ISIMIP2b utilizes
harmonized climate forcings derived from gridded, daily bias-adjusted climate data of various CMIP5 (5[th] coupled
model intercomparison project) global circulation models (GCMs) (Frieler et al., 2017; Lange, 2016) as well as with
the use of global annual atmospheric $CO_2$ concentration, and harmonized annual land use maps (Goldewijk et al.,
2017). The application of bias-corrected climate data ensures that the climate used by the land surface models is
consistent with observations over the last 40 years of the historical period. We compared the historical data
calculated by the different models with three five-year periods distributed over the XXI[st] century: the beginning
(2001-2005, called historical scenario), a middle scenario (2051-2055) and an end scenario (2091-2095). In order to
simulate mid and end century periods (2051-2055 and 2091-2095), we have used two century-scale scenarios called



Representative Concentration Pathway (RCP). These scenarios have been defined by the Intergovernmental Panel
on Climate Change (IPCC) (van Vuuren et al., 2011) and correspond to common socio-economic pathways followed
by the world's population. Here, we focused on RCP 2.6, which represents an active reduction of greenhouse gas
emissions to comply with the Paris Agreement, and RCP 6.0, which represents more or less *business as usual*. RCP
2.6 is predicted to produce a radiation forcing of 2.6 $W.m^{-2}$, whereas RCP 6.0 would result in a radiation forcing of
6 $W.m^{-2}$.
For each combination of LSMs (LPJmL or ORCHIDEE) and GCMs (CM5a or ESM2m), we calculated the mean
over 5 years at the beginning (2001 - 2005), mid (2051 - 2055) and end (2091 - 2095) of the $XXI^{st}$ century. The
cross scheme of two land surface models and two GCMs enabled us to establish whether estimations of runoff
are influenced more by rainfall projection or the representation of soil hydrologic characteristics. When
predictions will be driven by soil hydrologic properties, highest differences in runoff predictions are expected
between LPmL_(CM5a or ESM2m) and ORCHIDEE_(CM5a or ESM2m) projections than between LPJmL_CM5a and
ORCHIDEE_CM5a or between LMJmL_ESM2m and ORCHIDEE_ESM2m. Contrarily, when predictions will be driven
by rainfall projections, highest differences in runoff predictions are expected between LPmL_CM5a and
ORCHIDEE_CM5a or between LPJmL_ESM2m and ORCHIDEE-ESM2m projections than between LPJmL_CM5a and
LMJmL_ESM2m or between ORCHIDEE_CM5a and ORCHIDEE_ESM2m.

2.4. Statistical tests to assess AP and LP areas
AP or LP areas were assessed by comparing the $K_f$ and runoff values of each grid point with its corresponding
spatial median. Median runoff was computed for the whole of Europe for each five-year average period studied.
LP areas were characterized by low $K_f$ and high runoff, while AP areas were characterized by the opposite (see Eq.
(3a) and (3b)). We employed the classical approach described by Reimann et al., (2005) by classifying as outliers
values higher than the median +2 x (median average deviation) (MAD) or lower than the median -2 x MAD. Rather
than excluding data points as outliers, we identified data points with unusually high or low values, later referred
as anomalies. Thus, we used a threshold lower than 2 MAD for deviation definition and chose to fix a 1 MAD



threshold. MAD was computed as $median(|x_i| - median(x))$, $x$ being successively runoff and $K_f$ for the $i$ grid
points where $K_f$ can be estimated (see Eq. (3a) and (3b)).
For each combination of LSM (ORCHIDEE or LPJmL) x GCM (CM5a or ESM2m) and each time period (t=2001-2005;
2051-2055 or 2019-2095) with the two climate change scenarios (RCP 2.6 or RCP 6.0) applied for the periods 2051-
2055 and 2091-2095, we have defined LP and AP areas as follows:
•    Areas with soils exhibiting high potentiality of Cu leaching (LP areas) under 1 MAD threshold (named LP) for

193        a 5 years mean time period *t* were defined as areas where grid points *I* have:

$$\begin{cases} K_f(i) < Median\left(European\ K_f\right) - 1\ MAD\left(European\ K_f\right) \\ Runoff(t,i) > Median\left(European\ runoff\ (t) + 1\ MAD\left(European\ Runoff(t)\right)\right) \end{cases} \quad (3a)$$

•    Areas with soils exhibiting low potentiality of leaching corresponding to soils of high Cu accumulation potentiality

196        (AP areas) under 1 MAD threshold (named AP) for a 5 years mean time period *t* were defined as areas where grid

197        points *i* have:

$$\begin{cases} K_f(i) > Median\left(European\ K_f\right) + 1\ MAD\left(European\ K_f\right) \\ Runoff(t,i) < Median\left(European\ runoff\ (t) - 1\ MAD\left(European\ Runoff(t)\right)\right) \end{cases} \quad (3b)$$


The benefits of this approach is that it is not affected by the set of coefficients chosen to compute $K_f$, and it
removes the absolute nature of the values, but focus on the highest (and lowest) values. We used R 4.1.2 (R Core
Team, 2021) to compute anomalies and perform the figures.

3. Results
3.1. $K_f$ estimations at the European scale



The empirical equations extracted from our literature review to estimate $K_f$ are given in Table 1. We collected 15
equations allowing to calculate $K_f$ as the coefficient of partition between total Cu and Cu in solution. Among these
equations, pH was found the more decisive factor in $K_f$ estimation (8/15 relationships). Indeed, $K_f$ is positively
correlated to pH with a partial slope for pH around 0.3 for four of these eight relationships so that the more
alkaline the soil is, the highest the ratio total Cu/Cu in solution is. Soil organic matter (OM) or OC is less often a
parameter in the $K_f$ equations (4/15 relationships) but, when present, partial slope for OM/OC is higher than that
for pH. Three of the 4 papers concerned found a positive relationship between OM and $K_f$ while Mondaca et al.,
(2015) found a negative partial slope for soil OM or dissolved OC (Table 1, Eq. (12d)). However, this Eq. (12d)
concerns chilies soils and includes a positive partial slope for the CEC. The CEC value can be viewed as the sum of
clay and soil OM contents, so that the over whole partial slope of OM is compensated in that particular situation.





**Table 1.: Transfer functions reviewed from literature to estimate partition coefficient of Cu. R.V stand for response variable and Int. for intercept. Most studies fitted $K_f$ defined as $K_f = [Cu]_{soil}/[Cu]_{solution}^{n-opt}$ in $L.kg^{-1}$, $Cu_{soil}$ or Cu tot in $mg.kg^{-1}$, DOC (dissolved organic carbon) in $mg.L^{-1}$, OM (soil organic matter) in %, CEC in $cmol.kg^{-1}$, standard error around fitted coefficient are reported when indicated in the original article**

| Author | Eq | R.V | Int. | Log (Cu tot) | pH | Log (OM) | Log (DOC) | other | n-opt | R2 | number of data | Range Cu tot | Range OM | Range DOC | Range pH |
|---|---|---|---|---|---|---|---|---|---|---|---|---|---|---|---|
| (Vulkan et al., 2000) | 4 | Log ($K_f$) | 1.74 | | 0.34 | | -0.58 | | | 0.42 | 21 | 19-8645 | | 9.8-69.8 | 5.5-8 |
| (Sauvé et al., 2000) | 5a | Log ($K_f$) | 1.49 ±0.13 | | 0.27 ±0.02 | | | | | 0.29 | 447 | 6.8-82850 | | | |
| (Sauvé et al., 2000) | 5b | Log ($K_f$) | 1.75 ±0.12 | | 0.21 ±0.02 | 0.51 ±0.06 | | | | 0.42 | 353 | 6.8-82850 | | | |
| (Degryse et al., 2009) | 6a | Log ($K_f$) | 0.6 | | 0.37 | | | | | 0.34 | 129 | | | | |
| (Degryse et al., 2009) | 6b | Log ($K_f$) | 0.45 | | 0.34 | | | 0.65 log (CEC %) | | 0.44 | 128 | | | | |
| (Unamuno et al., 2009) | 7a | Log ($K_f$) | 1.95 | | 0.16 | | | | | 0.15 | 29 | 18-10389 | | | |
| (Unamuno et al., 2009) | 7b | Log ($K_f$) | 2.383 | 0.46 | | | | | | 0.61 | 29 | 18-10389 | | | |
| (Unamuno et al., 2009) | 7c | Log ($K_f$) | 1.99 | 0.42 | 0.06 | | | | | 0.63 | 29 | 18-10389 | | | |
| (Groenenberg et al., 2010) | 8a | Log ($K_f$) | 2.26 | | 0.89 | 0.9 | | | 0.85 | 0.87 | 216 | 0.1-326 | 2-97.8 | | 3.3-8.3 |
| (Ivezić et al., 2012) | 9a | Log ($K_f$) | 3.98 | | | 0.48 | -0.59 | | | 0.5 | 74 | 5.7-141 | | 0.9-10.2 | 4.3-8.1 |
| (Mondaca et al., 2015) | 10a | Log ($K_f$) | 1.05 | 0.7 | | -1.06 | | | | 0.46 | 86 | 56-4441 | 12.0-62 | | 6.2-7.8 |
| (Mondaca et al., 2015) | 10b | Log ($K_f$) | 2.88 | 0.41 | | | -1.03 | | | 0.77 | 86 | 56-4441 | 12.0-62 | | 6.2-7.8 |
| (Li et al., 2017) | 11a | Log ($K_f$) | 3.12 | 0.47 | | | -0.66 | | | 0.28 | 34 | | | | |
| (Li et al., 2017) | 11b | Log ($K_f$) | 2.179 | -0.45 * log (Cu solution) $\mu mol.L^{-1}$ | | | | | | 0.42 | 34 | | | | |
| (Li et al., 2017) | 11c | Log ($K_f$) | 2.59 | 0.617 | | | -1.55 | | | 0.88 | 20 | | | | |



Over the 15 equations, the estimation of $K_f$ according to Sauvé et al., (2000a) with Eq. (5a) or (5b) (Table 1) is the
most robust as determined over a wide range of soils (more than 400 points). The estimations are based on a
large gradient of in situ total soil Cu concentrations, even though the highest total soil Cu concentration is higher
than what was observed in Europe with the JRC's soil survey. The authors proposed two equations based on a
compilation of about 400 data points from long-term contaminated samples. One of the equations considers OM
values, whereas the other does not due to a lack of information in the gathered data. Finally, due to the well-
known importance in OM for binding with Cu, the Eq. (5b) was selected for our application at the Europe scale
and $K_f$ was calculated as following:
$$\log_{10}(K_f) = 1.75 + 0.21 \times pH + 0.51 \log_{10}(OM)$$
with $K_f$ in L.Kg$^{-1}$ and OM being the soil organic matter content calculated as OM = 2 x OC from JRC following
Pribyl (2010).
$K_f$ values display a range of 4600 to 21500 L.kg$^{-1}$ with a median value of  9829 L.kg$^{-1}$. $K_f$ values below 8000 L.kg$^{-1}$
and above 12000 L.kg$^{-1}$ respectively represent low and high anomalies for $K_f$. On the European scale, a
heterogeneous distribution can be seen when using equation (5b), as shown in (Fig. 1).



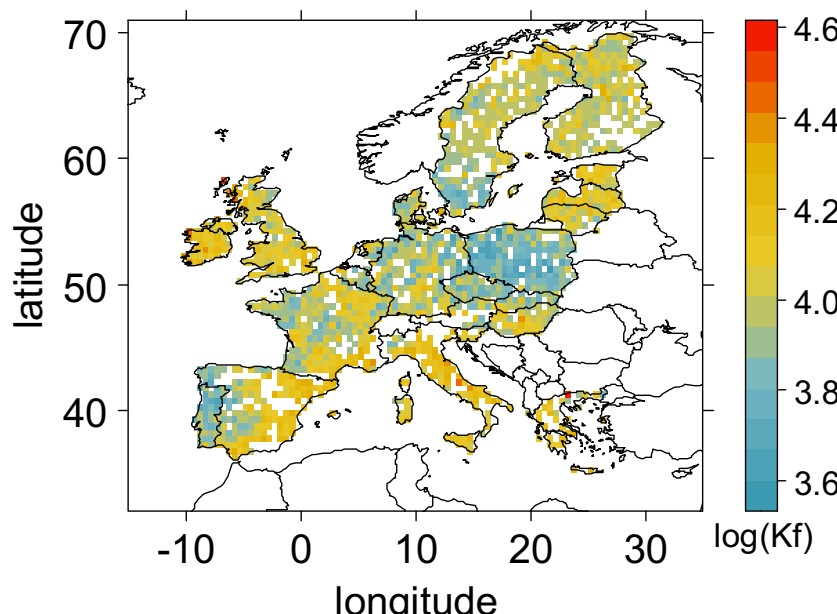

Fig. 1: Map of $\log_{10}(K_f)$ in Europe at 0.5° following Eq. (5b) applied to soil Cu contents. White pixels correspond

to pixel without OC measurement, then no $K_f$ calcul.

Beyond the EU's administrative borders (*e.g.* Switzerland and Norway), in certain mountain areas there is a lack

of OC data which isn't supplied by the JRC. Cu partitioning in soil solution is low around the Mediterranean, UK,

Baltic and Nordic regions with high $K_f$ (>12000 L.kg$^{-1}$). This accounts for 29.9 % of the grid cells, where deposited

Cu can thus accumulate in soils. On the contrary, high partition of Cu into soil solution can be found in 20.1% of

the grid cells where values of $K_f$ are low (<8000 L.kg$^{-1}$), thus providing soils with a tendency of acting as a source

of copper for other ecosystems, depending on the runoff. This occurs for instance near Portugal and Poland.

3.2. Modelling potential Cu leaching and accumulation in European soils at the beginning of the century (2001-

2005)



Over the two LSMs x 2 GCMs, the runoff values during the 2001-2005 period varied between 0 (LPJmL_CM5a
and LPJmL_ESM2m) and 5.4 mm.day$^{-1}$ (LPJmL_CM5a). The mean runoff value over the two LSMs x 2 GCMs is 1.1
(± 0.1 standard deviation) mm.day$^{-1}$ (data shown in Fig S1). For this period, the 1MAD threshold gives rather
similar low and high runoff anomalies between couples of LSMs x GCMs, below 0.6, 0.6, 0.7, 0.6 mm.day$^{-1}$ and
above 1.3, 1.2, 1.3 and 1.1 mm.day$^{-1}$ respectively for ORCHIDEE_CM5a, ORCHIDEE_ESM2m, LPJmL_CM5a and
LPJmL_ESM2m. In addition, respectively 21.7, 22.1, 20.2 and 21.1 % of the grid cells are low runoff anomalies and
28.2, 27.9, 29.8 and 28.9 % of the grid cells are high runoff anomalies (see Table S1).
Fig. 2 represents the LP and AP areas for the 2001-2005 period and for the different combinations of LSMs
and GCMs. The amount of grid cells concerned by LP and AP areas varied among the LSMs x GCMs combinations
(Fig. 3 with the historical scenario and Table S1). However, spatial patterns are well conserved with more
similarities between the same LSM than between the same GCM. Globally, LP areas are located mostly in Northern
Portugal with scattered points around France, Germany and Scandinavia while AP areas are mostly found in South
East of Spain, South-Adriatic coast of Italy and scattered points in Hungary. But, with the ORCHIDEE LSM, AP areas
in South Spain are larger, and LP areas in France and East Europe are more scattered than with the LPJmL LSM.



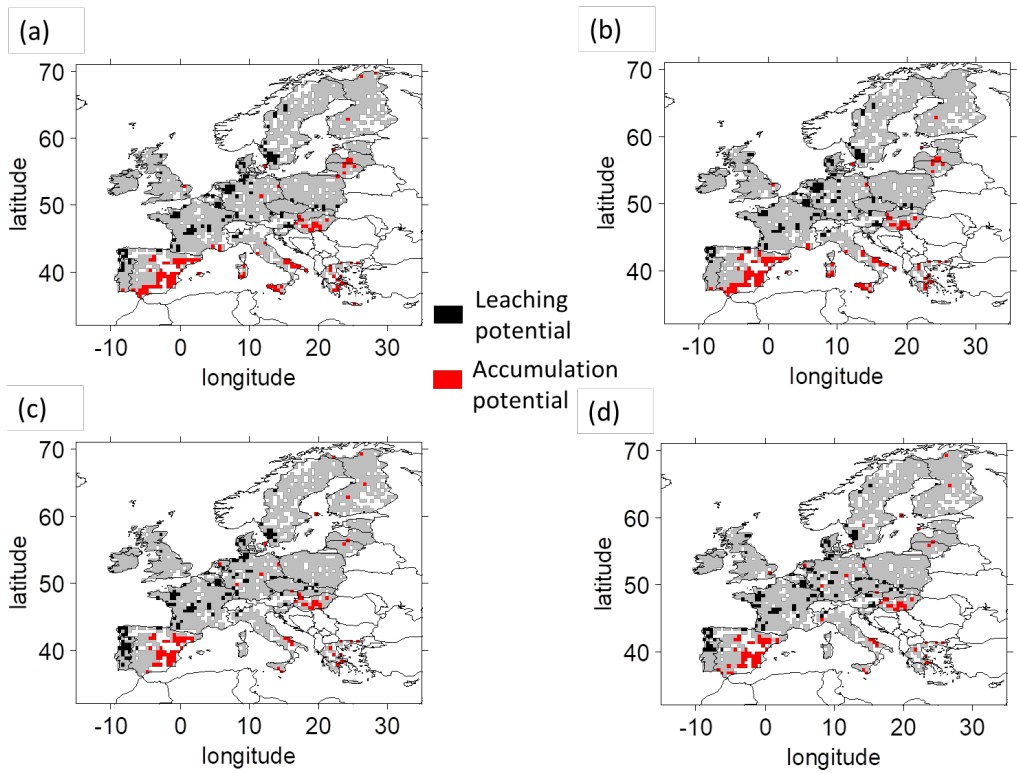


Fig. 2: Areas of potential for Cu leaching (LP) and accumulation (AP) over the historical (2001-2005) period for

the combinations of land surface scheme (ORCHIDEE in (a), (b) ; LPJmL in (c), (d)) and climate forcing (CM5a in (a),

(c) and ESM2m in (b), (d)). White pixels correspond to pixel without OC measurement, then no $K_f$ calcul.

Over the four combinations of LSMs and GCM, 6.4 ± 0.1 % (median, median deviation) of the grid cells are

concerned by LP (Fig. 3 (a)) and 6.7 ± 1.1 % by AP (Fig. 3(b)). Areas concerned by LP are almost equal between all

LSMs x GCMs even if ESM2m forcing leads to slightly less areas concerned by LP than CM5a. Much more AP areas

are predicted by ORCHIDEE LSM. LPJmL_CM5a combination has the smallest percentage of the grid cells

concerned by AP with 5.5 %, while ORCHIDEE_CM5A has the largest percentage with 8.0 % (Fig. 3(b)).

3.3. Modelling the evolutions of the LP areas over the century according to the different RCPs





For the two chosen climate change scenarios, median runoffs are expected to increase over the century for
the 2 LSMs x 2 GCMs combinations. For the mid 2051-2055 period, predicted runoff is 1.1 ± 0.1 mm.day$^{-1}$ with
RCP 2.6 and RCP 6.0 (mean, standard deviation of the 2 LSMs x 2 GCMs over the 5 years), (see Fig. S2 for RCP 2.6
and Fig. S4 for RCP 6.0). For the end 2091-2095 period, predicted runoff is also 1.1 ± 0.1 mm.day$^{-1}$ with RCP 2.6
but 1.0 ± 0.1 mm.day$^{-1}$ with RCP 6.0 (mean, standard deviation of the 2 LSMs x 2 GCMs over the 5 years), (Fig. S3
for RCP 2.6 and Fig. S5 for RCP 6.0). Table S1 shows that the amount of grid cells defined as high anomalies for
runoff tends to decrease by the end of the century while the amount of grid cells defined as low anomalies for
runoff tends to increase. However, tendencies for the 2051-2055 period are variable with in some cases an
increase or a decrease in percentage by comparison with the previous or subsequent periods (see Table S1).
Furthermore, among the different periods of climate change scenarios, the ratio of LP areas in percentage over
areas of high anomalies for runoff is not constant (see Table S1).
The evolution of areas in Europe concerned by LP for the different climate scenarios and the different LSMs x
GCMs combinations over the century is presented in percentage in Fig. 3(a). Compared to the historical values
and whatever the scenario, the median percentage of grid cells concerned by LP in 2091-2095 decreases by 1.2 ±
0.3 percentage points (median, median deviation) for RCP 2.6 and by 2.1 ± 0.5 percentage points for RCP 6.0.
Hence, at the end of the century, percentage of surfaces concerned by LP are 5.3 ± 0.3 % (median, median
deviation) for RCP 2.6 and 4.3 ± 0.6 % for RCP 6.0. Estimations of areas concerned by LP are relatively similar for
all the time period and climate change scenarios and for all LSMs x GCMs except ORCHIDEE_ESM2m that always
predicted smallest percentage of areas concerned by LP. Indeed, for ORCHIDEE_ESM2m the percentage of areas
concerned by LP are from 59% (RCP 6.0 2091-2095) to 79 % (RCP 6.0 2051-2055) smallest than the median
percentage of surfaces concerned by LP (see Fig. 3(a)).





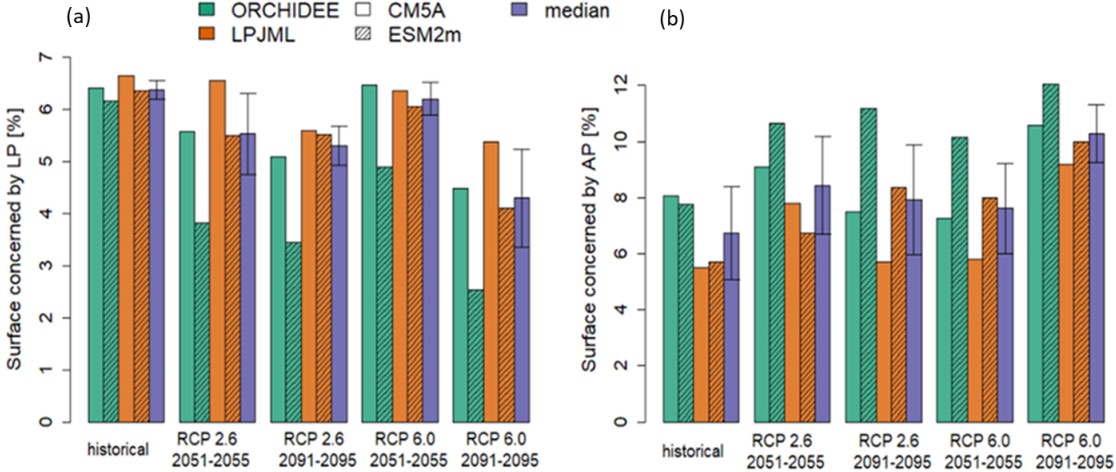


Fig. 3: Percentage of the grid cells concerned by Cu LP (a) and AP (b) for the different scenarios (historical=2001-2005, RCP 2.6 horizon 2050 and 2090 and RCP 6.0 horizon 2090). The 4 combinations of the 2 LSMs (ORCHIDEE in green and LPJmL in orange) and the 2 climate forcings (CM5a fill bars and ESM2m dashed bar) as well than median (purple) of the 4 models and median deviation (bar) are plotted.

The median evolution of LP during the century depends on the climate change scenario. With RCP 2.6, the median percentage of grid cells concerned by LP varied more between the historical scenario and the 2051-2055 one (-0.8 ± 0.4 percentage points, median, median deviation) than between the 2051-2055 and the 2091-2095 periods (-0.4 ± 0.3 percentage points). On the contrary, with RCP 6.0, the median percentage of grid cells concerned by LP areas decreases less from the historical scenario to the 2051-2055 one (-0.3 ± 0.2 percentage points, median, median deviation), than between the 2051-2055 and 2091-2095 periods (-2.0 ± 0.2 percentage points), see Fig. 3 (a). Furthermore, with RCP 2.6, estimations give 5.5 ± 0.5 % of the grid cells concerned by LP in 2051-2055 and 6.2 ± 0.2% with RCP 6.0, which is similar to the 2001-2005 estimate.

For all LSMs and GCMs and the two RCPs, LP areas mostly concern Portugal, north Germany and Scandinavia. In terms of LP risks, the combinations of GCMs and climate change scenarios mostly affect the quantity of dispersed





spots in East Europe and the southern extend of Portugal. By 2050, the decrease in LP areas mostly concerns
center of France, south of Portugal and north of Germany (Fig. 4 for the RCP 2.6 and Fig. 6 for the RCP 6.0). By
2090, the decrease in LP areas mostly concerns south of Portugal (Fig. 5 for the RCP 2.6 and Fig. 7 for the RCP 6.0).

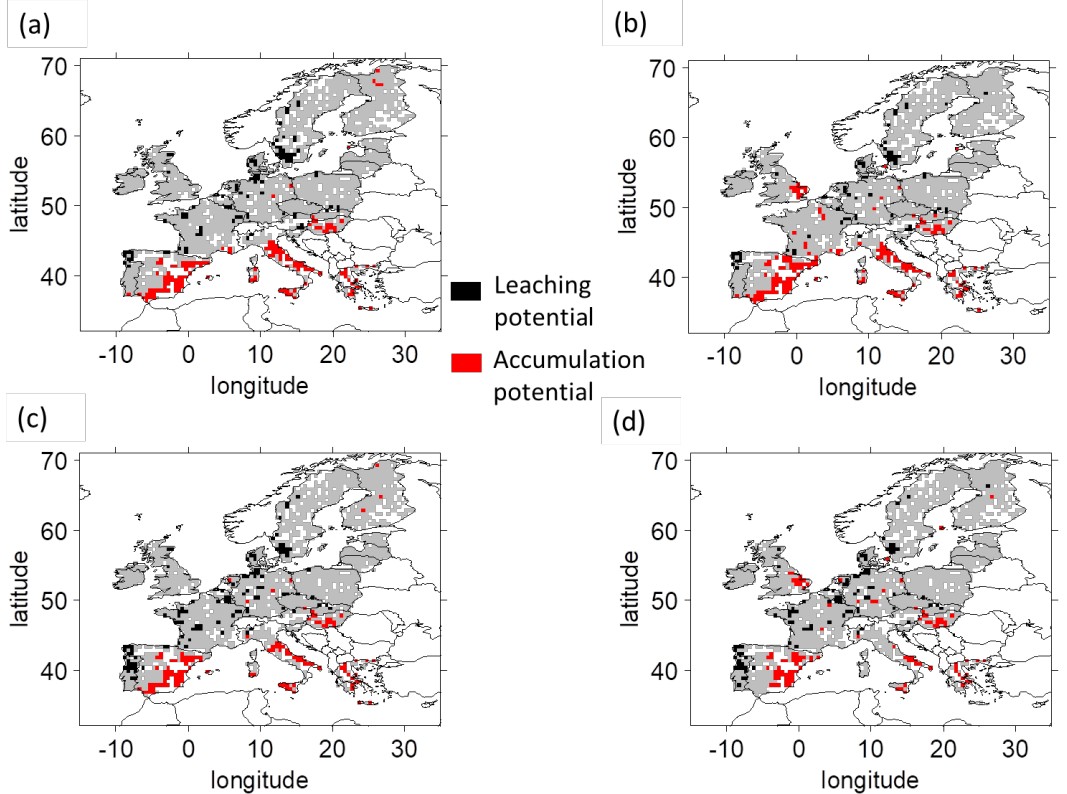


Fig. 4: Areas of potential for Cu leaching (LP) and accumulation (AP) over the RCP2.6 2051-2055 period for the
different combinations of land surface schemes (ORCHIDEE in (a), (b) ; LPJmL in (c), (d)) and climate forcings (CM5a
in (a), (c) and ESM2m in (b), (d)). White pixels correspond to pixel without OC measurement, then no $K_f$ calcul



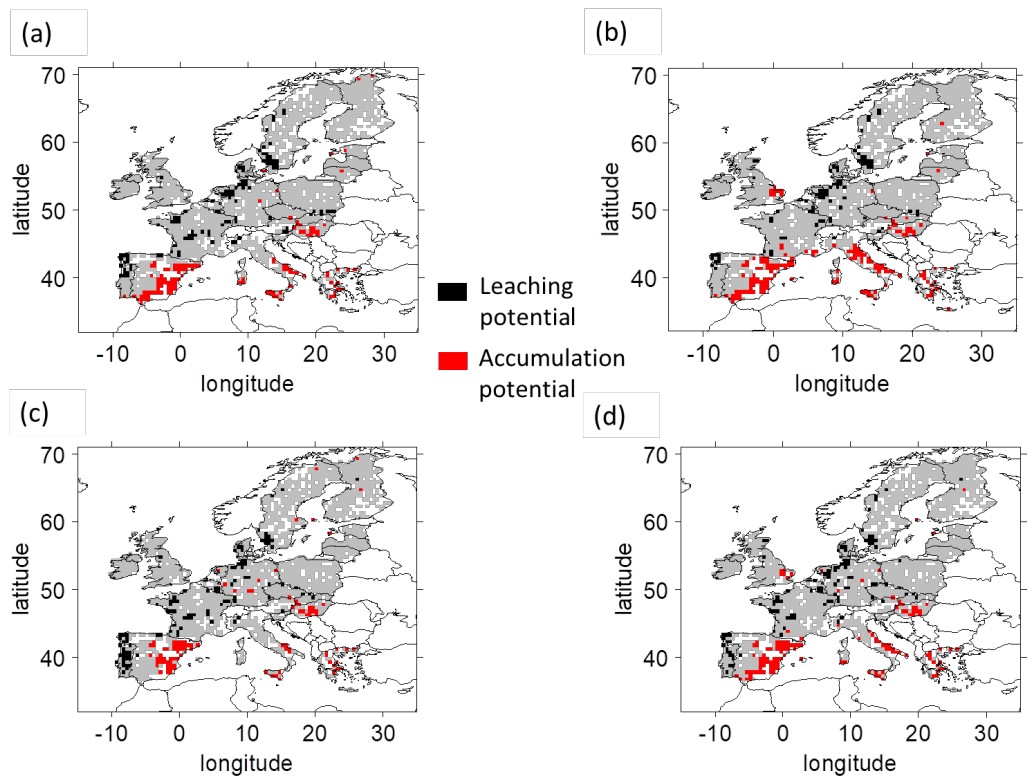


Fig. 5: Areas of potential for Cu leaching (LP) and accumulation (AP) over the RCP 2.6 2091-2095 period for the different combinations of land surface schemes (ORCHIDEE in (a), (b) ; LPJmL in (c), (d)) and climate forcings (CM5a in (a), (c) and ESM2m in (b), (d)). White pixels correspond to pixel without OC measurement, then no $K_f$ calcul.

3.4. Modelling the evolutions of the AP areas over the century according to the different RCPs

The evolution of areas in Europe concerned by AP for the different climate scenarios and the different LSMs x GCMs over the century is presented in percentage in Fig. 3(b). To the end of the century (2091-2095) and for the two climate change scenarios, the percentage of grid cells concerned by AP increases for all LSMs x GCMs except for ORCHIDEE_CM5a with RCP 2.6. AP area increases are highly variable between LSMs x GCMs, with a



smaller increase between historical period and 2091-2095 for RCP 2.6 than for RCP 6.0.
With RCP 2.6, and for all LSMs x GCMs, the percentage of surfaces concerned by AP increases between the
historical scenario and the mid-one (2051-2055). Between the mid and the 2091-2095 scenarios, the percentage
of grid cells concerned by AP increases for LSMs_ESM2m and decreases for LSMs_CM5a (see Fig. 3 (b)).
With RCP 6.0, the percentage of areas concerned by AP increases for all LSM x GCM except with ORCHIDEE_CM5a
between the historical scenario and the mid-one, and for all LSM x GCM combinations between the mid- and the
2091-2095 scenarios.
For all LSMs X GCMs and the two RCPs, AP areas are found in Sicilia, East Europe and South Spain. However, the
density and extent of the AP areas in these regions varied between LSMs x GCMs and climate change scenarios
(Fig. 4 and 5 for the RCP 2.6 respectively by 2050 and by 2090 and Fig. 6 and 7 for the RCP 6.0 respectively by
2050 and by 2090). Over the century, we found new AP areas in East Europe and Greece.



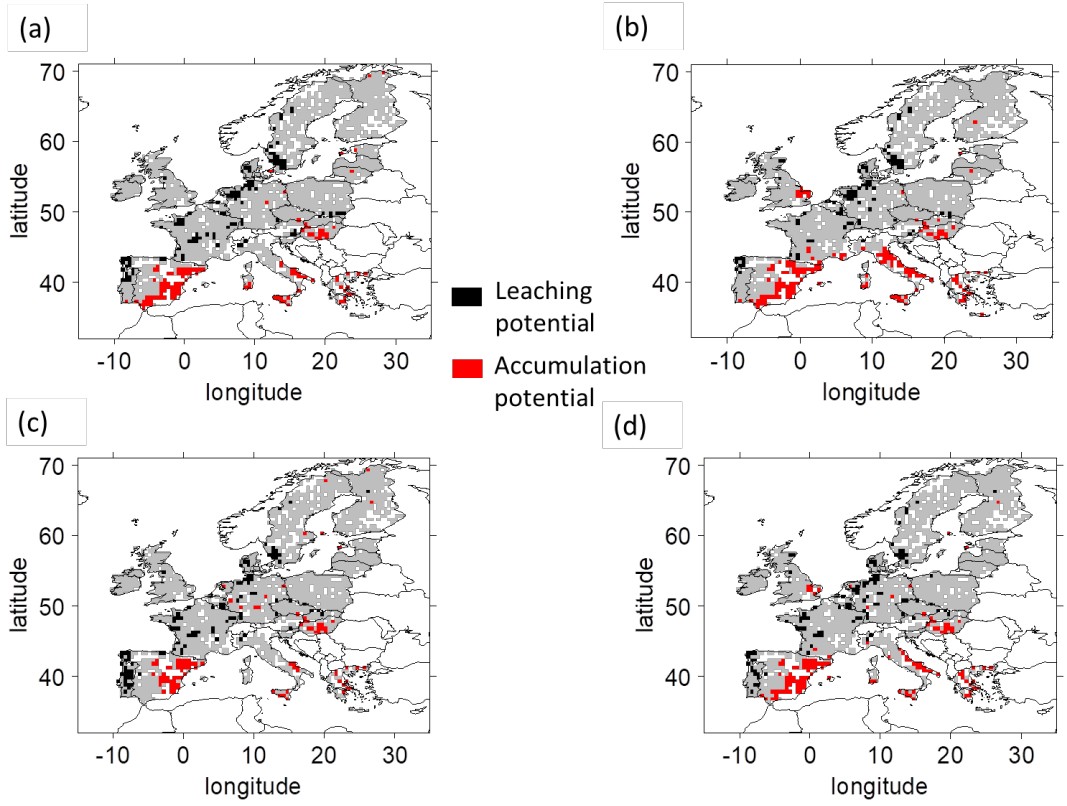


Fig. 6: Area of potential of leaching (LP) and accumulation (AP) over the RCP 6.0 2051-2055 period for the different

combination of land surface scheme (ORCHIDEE in (a), (b) ; LPJmL in (c), (d)) and climate forcings (CM5a in (a), (c)

and ESM2m in (b), (d)). White pixels correspond to pixel without OC measurement, then no $K_f$ calcul.




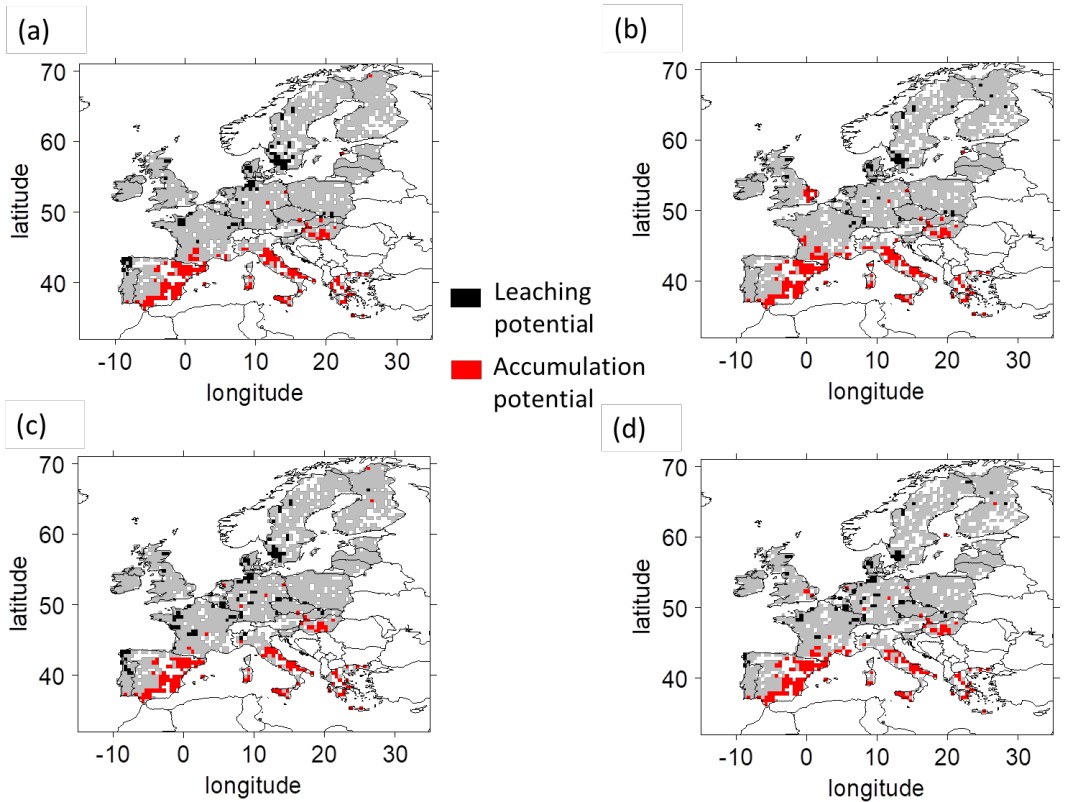


Fig. 7: Areas of Cu potential for leaching (LP) and accumulation (AP) potential over the RCP 6.0 2091-2095 period

for the different combinations of land surface schemes (ORCHIDEE in (a), (b) ; LPJmL in (c), (d)) and climate forcings

(CM5a in (a), (c) and ESM2m in (b), (d)). White pixels correspond to pixel without OC measurement, then no Kf

calcul.

Finally, over all LSMs x GCMs and climate change scenarios, the extent of areas of LP and AP in each region rather

depends on GCM than on LSM, with more similarities between ORCHIDEE_GCM (sub figures (a) and (b) in Fig. 2,

4, 5, 6, 7) and LPJmL_GCM (sub figures (c) and (d) in Figs. 2, 4, 5, 6, 7) than between LSM_CM5a (sub figures (a)

and (c) in Figs. 2, 4, 5, 6, 7) and LSM_ESM2m (sub figures (b) and (d) in Figs. 2, 4, 5, 6, 7).





## 4. Discussion


### 4.1. Modelling the role of copper sink or source with time for contaminated soils

This study aims at identifying potential leaching soil areas for Cu over Europe in order to identify locations where
soil may play a role in the Cu transfer from soil to aquatic ecosystems. To estimate the proportion of Cu reaching
soil solution, we chose to focus on the partitioning coefficient that considers soil properties. This specific choice
of $K_f$ coefficient rather than considering only the soil total Cu contents was made because Cu in solution is not
strictly correlated with total Cu, nor with other single soil properties as for instance pH and soil OM which are
both known to affect Cu partitioning and mobility. Thus, taking into account the variability of soil properties at
the European scale, the spatial distribution of Cu in solution was shown to be different from the spatial
distribution of total Cu (Sereni et al., 2022). Moreover, data on Cu in solution at large scales are not available
making impossible the estimation of AP or LP areas without using the $K_f$. Finally, the use of partition coefficient
allowed us to estimate risk areas without considering total soil Cu temporal variability and with the hypothesis
that pedological soil characteristics will not change at the time scale studied. This is a strong implicit assumption
but needed at that stage. Indeed even though some soil OM projections are available (Varney et al., 2022) to our
knowledge, future projections of pH values at European scale due to climate change are not available limiting our
capacities to calculate a time-dependent $K_f$. Furthermore, together with rainfall and soil moisture changes, climate
change is expected to also induce higher temperatures and shorter winters, so that a shift in cultures toward
North is expected (Hannah et al., 2013). Therefore, areas with currently low total soil Cu levels may potentially
experience a rise in Cu inputs from fungicides, which may subsequently be transported through freshwater
systems. Thus, the estimations of LP and AP as emphasized here, can be used to identify high-risk regions and
anticipate total content modifications that could occur with an eventual change in anthropogenic activities.
Indeed, land management changes due to land use changes or regulation changes may affect the use of Cu in
agriculture in the future with potential consequences on Cu leaching.



As a first step, the study conducted here could be used to highlight areas needing regulations to lower Cu input
thresholds. Indeed, the evolutions of the LP (and AP) areas we noticed are not only the reflection of the general
runoff evolution or of the current Cu risk but also underline areas of interest when combining risk linked to soil
contamination and climate change. For instance, in Eastern Europe low $K_f$ and high runoff result in Cu LP areas
with soils tending to act as source of Cu for the other ecosystems. However, in these cases, low amounts of total
soil Cu contents (Ballabio et al., 2018) limit the amount of Cu exports. In parallel, in Italy, we found high AP areas
whatever the modelling for at least one studied period and one RCP examined. In these vineyard regions, annual
Cu inputs are high, resulting in Cu accumulation in soil surface horizons with soils acting as a sink of Cu
contamination. These high total Cu concentrations could further enter the food web (García-Esparza et al., 2006)
or be exported with soil particles (Imfeld et al., 2020) due to rain erosion (El Azzi et al., 2013). Highly erosive storm
events predicted to increase during the next decades in Europe are another risk factor for freshwater
contamination even in AP areas. Hence, to go further on, localization of areas with exogenous risks of Cu
dissemination have to be identified to reinforce the predictions, e.g. by coupling studies of leaching potential as
the one we conducted here with erosion risk studies (Ballabio et al., 2017) and with retention pond localization
areas.

4.2. Temporal evolution of data and scope of the modelling analysis
To reduce intra and inter annual variability the modelling conducted here focused on 5-years means, thus aimed
at smoothing seasonal variability of runoff or of Cu inputs. The $K_f$ we calculated was not a dynamic value since we
did not make hypothesis about the temporal evolution of soil organic carbon or pH. Furthermore, $K_f$ is defined on
the assumption that there is equilibrium between the solid and solution phases. This means that the amount of
Cu in solution estimated by this method may be less than that present immediately after Cu application and before
equilibrium is reached (McBride et al., 1997). Despite all, our results showed a good agreement between the four
LSMs x GCMs in their projection of amount of grid cells concerned by both LP and AP, validating the use of their
median to perform projections in the absence of in situ validation. The scope of our predictions had limits that





rely on the difficulties to predict whether rain- and snow-falls and runoff will evolve in terms of intensity and
frequency, even if alternations of drying and rewetting events may affect Cu partitioning between phases
(Christensen and Christensen, 2003; Han et al., 2001). To gain field reality at the scale of territory for example,
modelling will require to account for the time periods of year with higher rain- and snowfalls amounts coinciding
with periods of Cu use, for instance in agriculture and vineyards (Banas et al., 2010; Ribolzi et al., 2002). Indeed,
if intense rainfall occurs close to Cu fungicide applications, a larger Cu amount than expected may be exported
through runoff. Thus, regional soil Cu budgets require the use of temporal model, which accounts for the regular
inputs and outputs of Cu from vegetation and runoff. Finally, the identification of the areas with high risks of soil
Cu leaching or accumulation we made in this study can be viewed as a first step for the risk evolution assessment
of Cu contamination useful for land management or Cu-fertilizer applications regulations.

5. Conclusion
Our approach to assess European areas with a potential to accumulate or leach copper from soils was not
straightforward but included several steps. We focused first on the means to calculate Cu partitioning. By
reviewing existing Cu $K_f$'s equations we pointed out pH and soil OM contents as important determinants and more
precisely that the OM partial effect was larger than the pH one. Then, using the European maps of soil
characteristic data, we computed the map of $K_f$ at the 0.5° scale, highlighting areas with high risk to leach or to
accumulate Cu for a given soil's overall content or upcoming. The estimation of LP and AP areas for current and
future soil runoffs under two RCPs with couples of two GCMs x 2 LSMs was thereafter performed by comparing
anomalies for both $K_f$ and runoffs. Interestingly, our first result showed that the variations in the number of LP
and AP grid points was not only due to variations in the runoff intensities distribution but also to their localization.
Indeed, the ratio of percentage of areas of LP or AP over areas of high or low anomalies was not constant during
the century. At the beginning of the XXI[st] century our study showed that comparable amounts of grid cells are
concerned by LP and by AP (between [6.2% - 6.4%] and between [5.5% - 8.0%] respectively). During the century,
AP areas were found to increase for all the LSMs x GCMs and the two RCPs. On the contrary, for the two RCPs and



three over the four LSMs x GCMs, LP areas were found to decrease during the century compared to the current
estimation. Projections for 2090 with RCP 2.6 considering median (and median deviation), indicated 5.3 ± 0.3% of
the grid cells concerned by LP areas and 7.9 ± 1.3% by AP areas. Projections for 2090 with RCP 6.0 showed a
slightly smallest amount of grid cells concerned by LP and a highest by AP with respectively 4.3 ± 0.6% and 10.3 ±
0.7 % of the grid cells (median, median deviation). Surprisingly, the total amount of grid cells concerned by the
two risks of AP and LP is rather similar between the two climate change scenarios with estimation between 13.2
± 1.3 and 14.6 ± 1.3%. This was due, however, to opposite trends in the evolution of LP and AP areas. Their relative
proportions and period of main variations differed with most of the evolution between historical period and 2051-
2055 for RCP 2.6 and between 2051-2055 and 2091-2095 for RCP 6.0. Finally, we showed that even if the number
of grid points identified with LP and AP may varied between LSMs x GCMs models and climate change scenarios,
their localizations are roughly conserved, emphasizing the necessity to precise monitoring in Cu application on
these areas.

Code availability:
The code can be provided upon request.

Data availability:
The data can be provided upon request

Credit authorships contribution statement:
Laura Sereni: Methodology, Formal analysis, Data processing, Writing original draft.
Julie-Mai Paris: Formal analysis, Initial data processing, Writing original draft.
Isabelle Lamy: Methodology Conceptualization, Writing review and editing, Supervision, Funding acquisition





Bertrand Guenet: Methodology, conceptualization, Writing review and editing, supervision, Project administration,

Declaration of competing interests
The authors declare that they have no known competing financial interests or personal relationships that could
have appeared to influence the work reported in this paper.

Acknowledgments
Parts of this study were financially supported by the Labex BASC through the Connexion project. LS thanks the Ecole
Normale Supérieure (ENS) for funding her PhD. The authors thank Nathalie de Noblet-Ducoudré for valuable
discussions on this paper.

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
