# Peer review of "Estimations of soil metal accumulation or leaching potentials under climate change scenarios: the Example of"

_EGUsphere, 2023_

## Author Response (AR1)

**Responses to reviewer 1**
**General comments**

*We thank the reviewer for the constructive evaluation of the manuscript. Please find below our answers to questions/comments. Comments from the reviewer were left intentionally in this document and written in roman font. Our answers are written in italics.*

This manuscript describes the results of a modelling study to the leaching and accumulation potential of copper in soils. This study was performed at a European scale at a 0.5degree model resolution. The authors pursue an original concept in the assessment of the leaching and accumulation potential and is therefore interesting for the audience of Soil. However, the manuscript contains several issues that need to be addressed in a revised version of the manuscript. In the specific comments below, I will list and explain these major issues.

*Thank you for the positive comment. The revised version of the manuscript has been improved including your suggestions and detailed according to your comments. We hope that the explanations below will answer your concerns.*

**Specific comments**
**Title**

The title suggests that the mobility of metals, in this case copper, is simulated. However, this is not the case. This study only assessed the leaching potential (LP and accumulation potential (AP) of copper in European soils and this should be clear from the title.

*In accordance with this comment, we changed for a new title to emphasize that the study focused on potentials risks:*
*Estimations of soil metal accumulation or leaching potentials under climate change scenarios: the Example of copper on a European scale*

**Definitions**

In the introduction section, the terms "leaching" and "runoff" are not clearly defined, which may cause confusion for the reader. In land surface models such as used in this study, runoff consists of the lateral export of water via surface pathways from each model element / grid cell (streamflow and, but less important at larger scales, overland flow). Leaching refers to the vertical transport from the topsoil or root zone via percolating water.
Assuming that the contribution of overland flow to runoff is negligible compared to the contribution of streamflow/river discharge, the use of runoff data as an indicator for leaching of substances from the topsoil can be justified. Despite leaching and runoff are related, they are not the same. Leaching is not caused by local runoff processes as line 83 suggests.

*We agree with these definitions. We actually used the term leaching as the desorption of Cu from soil phases and its transport within rain flow. We rephrased the lines l86-90 to avoid such a confusion. This now reads " i) estimate the areas the most likely to lose soil Cu within soil solution and waterflows, thereafter named leaching potential areas [ LP], for the*

*beginning of the century and ii) predict their changes according to different climate change scenarios".*

Likewise, use of the terms "sinks" and "sources" may lead to confusion. The question is which sinks and sources are meant by the authors. Total copper amount in the soil system? Copper in the soil solution? Copper adsorbed to soil? It looks like that the authors refer to the sinks and sources of adsorbed copper in soil. This should be clearly stated in the manuscript, but it is perhaps even better to avoid these terms.

*We didn't notice the confusion. As suggested, we found it better to suppress these terms. This allows to shorten the sentence without changing its meaning. Also, there is no mention of source or sink in the manuscript*

Furthermore, the authors link both the LP areas (i.e. areas with relatively low Kf and high runoff values) and AP areas (i.e. areas with relatively high Kf and low runoff values) to risks without a clear definition of the associated risks. In the discussion sections, the areas with relatively high Kf and high runoff values are associated with erosion risk (see also comment below). It is not logical to me why the areas with intermediate values for runoff and kF are not at risk (in other words: why it is considered to be a risk when the vast majority of copper is accumulated in soil or the vast majority of the copper is leached, but not when part of the copper is leached and the other part is accumulated). This needs further clarification. Alternatively avoid the use of risk in this context or clearly define which specific risks are associated with the LP and AP areas.

*The term "risks" is always subject to discussions as it depends on the references and end points chosen as well as on contexts. Therefore, we rather decided to focus on the "potential risk" (e.g. the most likely to leach or accumulate) with the aim to highlight the areas that require further consideration and local risk assessment. The lines l86-90 now reads " i) estimate the areas the most likely to lose soil Cu within soil solution and waterflows, thereafter named leaching potential areas [ LP], for the beginning of the century and ii) predict their changes according to different climate change scenarios. Additionally, we aimed to estimate the areas the most likely to accumulate Cu, thereafter named accumulation potential areas [AP] "*

**Datasets of copper concentrations in soil**

In line 85 the authors state that there is a lack of copper concentrations in soil to substantiate the choice to calculate the leaching and accumulation potential only on predicted copper partition coefficients and runoff and not on actual (or future) copper concentrations. However, despite the authors refer to the main authors of existing datasets on current copper concentrations in the European topsoil, namely the Foregs database (Salminen et al.) and the LUCAS topsoil survey (Ballabio et al.), they seem to neglect the existence of these datasets. The authors should justify why they did not use these datasets in this study.

*To our knowledge the Foregs and LUCAS dataset provide actual total Cu content but no information about the future deposition of Cu that largely depends on practices and policies. Studies aiming at estimating the future concentration of Cu in soil. For instance, Droz et al.*

*(2021) had to make strong hypothesis concerning Cu application (for instance at the maximal level authorized by European policies). Here, we wanted to avoid such a strong hypothesis on the application rate but rather to highlight the risks and consequences of application that differ from place to place. The lines 93 now reads "Due to the lack of information about the future Cu deposition whatever its form, we developed a method using the partition coefficient ($K_f$) at the equilibrium between solid and solution phases to determine areas with high or low potential of leaching whatever total Cu concentration. Regarding the lack of data about future deposited amounts at large scale, using $K_f$ was necessary to estimate the Cu mobility potential"*

**Determination of LP and AP areas**

Although not explicitly stated in the manuscript, the median and median average deviation (MAD) values for runoff seems to have been determined for each time period and combination of GCM and LSM. The reason for this choice is given, but this has implications for the interpretation of the results for the climate projections. The current way of presenting the leaching potential (LP) and accumulation potential (AP) areas under the climate change scenarios do not allow a direct comparison with the present situation in terms of a temporal increase or decrease in surface area with leaching or accumulation potential. For this the projected LP and AP areas should be calculated using the present or historic median and MAD values for runoff. The current way of presenting may also explain the relatively small changes in surface area for the different scenarios in figure 3

*Indeed, the choice of the reference to calculate deviation needs to make a strong assumption. We choose to calculate the MAD to each time period to emphasize the spatial deviation. When considering the actual rainfall regime as a reference, we consider that the historical environmental risk assessment well considers the spatial risk variability. Here, we rather consider that this spatial environmental variability has to be taken into account to define environmental policies.*

*Furthermore, it has been shown that the sensitivity of organisms to contamination depends on the historical climate encountered (Kwon et al., 2013 https://doi.org/10.1016/j.soilbio.2013.01.023, 2013.; Evans and Wallenstein, 2012 https://doi.org/10.1007/s10533-011-9638-3, 2012; Sereni et al., 2022 https://doi.org/10.1007/s11356-022-19093-2.) Taking into account the historical rainfall regime is also of major interest but requires other studies.*
*The lines 222-225 changed to specify this and now read:" We choose to calculate the MAD to each time period to emphasize the spatial variability. Anomalies identification could also be done using the historical runoff as a reference and looking at its change with time. However, when considering the actual rainfall regime as a reference, the current environmental risk well considers the spatial risk variability.*

In addition, considering the 5-year time period for which the runoff was averaged, it could be questioned what the added value of using a LSMs to the determination of the mean runoff values compared to a simple subtraction of evapotranspiration from precipitation.

*The added value lies in the fact that runoff also depends on the hydrological soil scheme provided by the LSM. This is why we cross the results from the GCM to the LSM and why the results differ between each couple. This has been detailed in lines 183-185 that now read: "The cross scheme of two land surface models and two GCMs enabled us to establish whether estimations of runoff are influenced by rainfall projection provided by the GCMs or the representation of soil hydrologic characteristics provided by the LSMs."*

**Discussion**

Instead of discussing the results of this study, the current discussion contains too many obvious statements that are not based on own results, but on other studies or common knowledge, for example, the discussion on the effects of processes and factors occurring on much smaller temporal and spatial scales than considered in this study (lines 183-193).

In addition, in the preceding section 4.1, the discussion the risk associated with erosion of copper-contaminated soil is confusing (lines 155-173). First, the areas with relatively large 5-year averaged runoff do - to my knowledge - not necessarily coincide with areas with high soil erosion risk. Average runoff is driven by precipitation and evapotranspiration, whereas soil erosion is driven by high-intensity rainfall events, erodible soils and sloping areas. But even if the areas of high runoff would coincide with those of high erosion risk, the question arises why this has not been considered as a third class of areas with erosion potential (areas with of large values for both runoff values and kf).

*You are right, the areas with large or little precipitations do not correspond to areas with soil erosion risks, but this represents a third class of risk. This has not been included in this study for different reasons. First, a 3 factorial study appears to be confusing without the first identification of areas at risk for the 2 factorial studies conducted here. Secondly, the erosion-transport processes are not at the same spatial nor at the same temporal scales as those studied here. Finally, the erosion data recently published were not included in this study but their use was suggested. We emphasized this difficult scale and rephrased lines 432-436 "Highly erosive storm events predicted to increase during the next decades in Europe are another risk factor for freshwater contamination even in AP areas, but are often very punctual and local […] e.g. by coupling studies of leaching potential as the one we conducted here with erosion risk studies (Panagos et al., 2021)"*

Related to a discission of the own results of the study, I miss an attempt to interpret and explain the drivers for the temporal changes in AP and LP areas (changes in rainfall, evapotranspiration, or both?). Moreover, I miss a discussion about the effects of possible climate-induced changes in soil pH and organic matter content to the LP and AP.

*Indeed, we did not discuss the drivers for runoff changes that would have required to produce new results by going into the output of the different models. We rather considered four models as a safety range and discussed whether the temporal trends are driven by the GCM or the LSM. Nevertheless, we add some sentences in the discussion to discuss more in details our results.*

*Considering the potential changes in pH or organic matter content, this is a tricky question as especially OM content is subject to change across the century according to both climate change and willing to increase soil organic C to reduce atmospheric C. Also, crossing climate change to soil organic content and pH evolution to assess contamination evolution would be a very different study. However, we precise in line 411-414 this difficulty by writing:" In particular, there are large uncertainties about the C stocks that may change as a result of climate change and dedicated policies for increasing the C stocks (Bruni et al., 2022). Besides, organic fertilizers applied to increase C stocks can change both pH and soil Cu content leading to supplementary uncertainties (Laurent et al., 2020). »*

**Conclusions**

The current conclusions section is predominantly a summarizing repletion of detailed results of the study. Many details can be omitted, and I would like to challenge the authors to formulate their conclusions at a higher level of abstraction. Furthermore, conclusions from the above new discussion points could also be added.

*According to your remarks, the conclusion has been shortened with removing specificity to this study. We reformulated some of the main generalizable results: "We hence provided a new method to emphasize at the regional scale the combined risk of both climate change and contamination. We pointed out that despite similar We pointed out that despite similar projections for the end of the 21st century, the trend during the century depends on the climate change scenario." (lines 472-474) And "We highlighted the areas of particular risk for application of Cu, emphasizing the necessity to precise monitoring in Cu application on these areas. Future studies would be gained in precision by taking into account the change of partitioning coefficient with soil variables or scenarios of Cu application taking into account the various forms (e.g., mineral or organic fungicides). » (lines 494-497)*

**Technical corrections**
*The modification suggested have been introduced in the main text and the rephrasing asked are detailed below in italic*

Pages 1-12
l. 11: "Soil contaminant deposition": please rephrase (contaminant inputs to soil?)

*Done*

l. 13-14: "leached through runoff": please rephrase as runoff does not cause laching (see specific comments above)

*This has been rephrased as: "rather be transported with runoff or accumulated"*

l. 18: "among contaminant": please remove
*Done*

l. 23: "XXIth" change into "21st" (please also change other occurrences of XXIst throughout the manuscript

*Done*

l. 24: "Grid point" : I expect that the model predictions and soil data refer to grid cells rather than points (but please check!), therefore use "grid cells " consistently throughout the manuscript.
*Done*

l. 48: remove the comma after "used"
*Done*

l. 65: define source and sink (see also specific comments)
*We deleted these terms to avoid ambiguities*

l. 65: "This knowledge": please rephrase (it is not clear to what "this" refers)
*This has been rephrased as: "Know and predict this leaching or retention, however, could"*

l. 70: "projections forecast": please rephrase (projections do not forecast) (an increase in rainfall and snowfall events is projected…)

*This has been rephrased as: "For instance, an increase in rain- and snow-fall events in winter in Northern Europe but a decrease in summer in the Mediterranean region are projected"*

l. 77: "the relationships between these changes in runoff and fluxes of elements is still poorly predicted": please rephrase

*This has been rephrased as: "However, predicting how these runoff changes will relate to elemental fluxes in the coming decades remain difficult.*

l. 79-81; "Thereafter named": please remove and put LP and AP between brackets
*Done*

l. 113: define Cu_total and Cu_solution and provide dimensions or units
*Done*

l. 140-141: put references to websites in the reference list (do this also for other references to websites). See instructions for submission on the SOIL website
*Done*

l. 147: "To estimate changes in runoff across century and to reduce uncertainties" this sentence needs clarification; please reformulate "across century"
*This has been rephrased as: "To estimate changes in soil runoff during the 21st century"*

l. 161: "have used" = "used"
*Done*

l. 172: "will be" = "are"
*The paragraph has been rephrased (see below)*

l. 173: "than": please rephrase, since this is not a comparison

*The paragraph has been rephrased (see below)*

l. 174: "will be" = "are"
*Done*

l. 176: "than": please rephrase, since this is not a comparison
*The paragraph has been rephrased (see below)*

l. 172-177: these sentences are very difficult to read because of the numerous abbreviations of model scenario runs. Please consider rephrasing by breaking the long sentences up into shorter sentences.

*These two sentences have been rephrased as :" When predictions are driven by soil hydrologic properties, highest differences in runoff predictions are expected between the couples of models with the same LSM but different GCM (e.g. for instance LPJmL_CM5a is closest to LPJmL_ESM2m than to ORCHIDEE_CM5a) Contrarily, when predictions are driven by rainfall projections, highest differences in runoff predictions are expected between couples of models with the same GCM but different LSM (e.g. for instance LPJmL_CM5a is closest to ORCHIDEE_CM5a than to LPJml_ESM2m)"*

l. 179: Please remove "statistical test" from the title of this subsection, as there is no statistical test mentioned in this subsection.
*Done*

l. 183-184 I wonder whether these sentences are still relevant, since the identification of outliers using this classification is not used for further analysis.
*This has been removed*

l. 185: 'data points" use other word or even better: remove entire subordinate clause
*This has been replaced by "grid cell"*

l. 186: "chose to fix a 1 MAD" please rephrase
*This has been rephrased as: We identified grid cells with unusually high or low values, later referred as anomalies, as data points above or below a 1 MAD deviation*

1. 200: "it is not affected by the set of coefficients chosen to compute Kf" : it is not clear to what "it" refers. Furthermore, I question whether this statement is true. I could argue that the identification of LP and AP areas is not very sensitive to the parameter values used for the calculation of Kf, but at the same time I could argue that it is not entirely insensitive to the parameter values (for example, thins of the effect of setting one of the parameter values regression coefficients to zero
*This has been rephrased as: "The benefit of this approach is that outliers identification is not affected by". Your remark is true, however the statement came from having noticed that*

*between all transfer functions the ratio between the coefficients of the different parameters is very often similar.*

l. 201: "but focus on the highest (and lowest) values": this subordinate clause can be removed
*This has been rephrased as: "but focus on the deviation to median."*

l. 210-211: "so that the more alkaline the soil is, the highest the ratio total Cu/Cu in solution is" Apart from the sloppy formulation, this statement is only true because of the positive correlation with pH (or, more precisely, the positive regression coefficient for pH) and not because of a regression coefficient of near 0.3. I would remove this subordinate clause
*Done*

Table 1: what does the parameter "n-opt' mean? The majority of the cells in this column are empty. Does this mean that the n-opt value is 1in these cases?
*You are true in this case n-opt is 1. But, it rather came from the fitting of the transfer function with a n-opt =1. This has been specified*

Pages 13- 28 (the line numberings starts at 1 again from page 13)
l. 5 :"JRC's soil survey": please provide a reference. Do "the authors" in the next sentence refer to the authors of this JRc's soil survey?
*This has been clarified.*

l. 10 : remove "×" from the equation
*Done*

l. 53: why median runoff? In the next sentences the mean runoff and standard deviation is given of the different combinations of GCM/LSM runs. I think this can be generalised to "runoff is projected to increase"
*The median is calculated per model while the mean is calculated over the 4 models. We precised at the end of the material and methods line 227-228 "In the next sections the results of temporal trends are presented using median per model and mean over the 4 models."*

l 64: "evolution" I would recommend to avoid the term "evolution" in this context. Replace by "change", also for other occurrences of this term throughout the manuscript
*Done*

l. 140: "partitioning coefficient that considers soil properties" please rephrase ("partitioning coefficient, which is calculated using pH and soil organic matter content")
*This has been rephrased as: "we chose to focus on the partitioning coefficient which is calculated based on soil properties (pH and OM here) others than total soil Cu"*

l. 140-145: I do not follow the reasoning here: the inference that taking into account the variability of soil properties at the European scale, the spatial distribution of Cu in solution was shown to be different from the spatial distribution of total Cu does not follow from the preceding sentences. Please rephrase.
*The previous sentence has been rephrased to specify*

l. 145-146: Again, the reasoning here is false because the dissolved Cu concentration in the soil solution does not represent the leaching or accumulation potential, but only on the actual leaching rates when the infiltration/percolation rates in soil are also know. Please rephrase
*This has been rephrased as: "However, data on Cu in solution at large scales are not available making impossible the direct estimation of transport within soil solution and of AP or LP areas without using the Kf"*

l. 155: "emphasised": please use other word
*Done*

l. 155: avoid the term "high-risk" (see also specific comments)
*Done*

l. 165: "these vineyard regions": please provide information/confirmation that the AP areas in Italy are indeed vineyard areas.
*Regions named have been specified*

l. 172-173: the coupling of studies of "retention pond localisation areas" requires further clarification (or remove the reference to such studies)
*This has been rephrased by "outlet characteristics".*

l. 177: "smoothing of" "Cu-inputs": I question this statement since the approach taken in this study to identify AP and LP areas is independent from Cu-inputs or Cu concentrations in soil

*The mention of Cu-inputs smoothing has been removed here and mentioned later in the discussion as:" Thus, local soil Cu budgets require the use of temporal model, which accounts for the regular inputs and outputs of Cu from vegetation and runoff but cannot be accounting with multiyear mean. »*

l. 186: what is meant by "scale of territory"? Please clarify
*This has been rephrased as: "at the local scale (here up to 50 km) such as landscape or catchment"*

l. 189: "a larger Cu amount than expected may be exported": what is the expected amount of exported Cu? In this study this was not quantified (see also specific comment about the manuscript title)
*This has been rephrased as: " a larger Cu amount than locally computed taking into account total Cu and Kf may be exported through runoff"*

l. 199: "the OM partial effect was larger than the pH one". This conclusion is new and not reported in the results section. Please remove or, if relevant (which I think it is), add a paragraph in the results section in which the partial effects of the soil properties on the Kf values is quantified.

*This was mentioned in the result section and has been detailed in: "partial slope for OM/OC is higher than that for pH which means that a small variation in soil OM content affects more Cu partitioning than a small variation in pH"*

l. 203-204: "Interestingly, our first result showed that the variations in the number of LP and AP grid points was not only due to variations in the runoff intensities distribution but also to their localization": What is meant by localization? Why the variations in the number of grid cells? Please clarify this sentence (and avoid the term "grid points").
*This has been removed from the conclusion according to your remark of simplification*

l. 213-215: "Surprisingly, the total amount of grid cells concerned by the two risks of AP and LP is rather similar between the two climate change scenarios with estimation between 13.2 ± 1.3 and 14.6 ± 1.3%": This sentence is ambiguous since it is not clear whether the number of AP and LP grid cells is similar or the number of AP and LP grid cells together is similar for the two climate change scenarios. Please rephrase (avoid the term "risk" (there is no risk of the accumulation potential!) and replace "amount" by "number")
*This has been rephrased in "Surprisingly, the total number of grid cells concerned by AP and LP estimated at the end of the century is rather similar, with estimation between 13.2 ± 1.3 (RCP 2.6) and 14.6 ± 1.3% (RCP 6.0). This was due, however, to opposite trends in the change of LP that decreases and AP areas that increase during the century."*

**Responses to reviewer 2**
*We thank the reviewer for the constructive evaluation of the manuscript. Please find below our answers to questions/comments. Comments from the reviewer were left intentionally in this document and written in roman font. Our answers are written in italics.*

This manuscript is well written and it models the evolution of copper partition coefficient changes in response to climate change (under two scenarios) for 3 periods (~2005 ; 2050 ; 2100) and with two different models (one soil explicit model, the other not). The regional approach appears to be well conducted but, to me, the title and the introduction of the manuscript are really misleading and the discussion lacks to address important points (regarding the limitations of the study and the lessons learnt from it).
*Thanks for your comments. The title has been changed in "Estimations of soil metal accumulation or leaching potentials under climate change scenarios: the Example of copper on a European scale". Besides, several sentences of the introduction have been rephrased lines 83-84:" However, predicting how these runoff changes will relate to elemental contaminant fluxes in the coming decades remains difficult." ; lines 86-90: "In this framework , our aim was twofold: i) estimate the areas the most likely to lose soil Cu within soil solution and waterflows, thereafter named leaching potential areas [LP], for the historical period (2001-2005) and ii) predict their changes according to different climate change scenarios. Additionally, we aimed to estimate the areas the most likely to accumulate Cu, thereafter named accumulation potential areas [AP]" ; line 106-108 "The rainfall predictions were analyzed at the 0.5° that is a common scale for land surface models allowing a multi-comparison to capture the variability in soil properties and rainfall regime."*

Overall, to the best of my understanding, only the potential to accumulate or release Cu in/from soils is modelled here, without taking the overall Cu concentration or the exposition risk of given soils to Cu contamination into account. I understand that the authors justify this by the fact that the Cu partition coefficient in soils does not depend on the total Cu concentrations but, still, it is fundamental in my opinion to relate this behaviour to the actual exposition of soils to Cu contamination, otherwise the maps may be totally misleading (e.g. soils from these regions have a very high potential to accumulate Cu >> OK, but this does not matter somehow if the Cu background level is low and soils are not exposed to anthropogenic Cu releases?)

*Thanks for your comments. Indeed, we aimed at assessing a map highlighting the areas with most concerns for contamination in a future climate. Considering that the exposition to Cu largely depends on anthropogenic activities that can change very fast, we argued that it was primordial to assess the potential risk first, and then to assess the risk posed by the scenario of inputs in a second time, which was not the purpose of this study. This has been detailed lines 499-501 "Future studies would gain in precision by taking into account the change of partitioning coefficient with soil change or scenarios of Cu application taking into account the various forms (e.g., mineral or organic fungicides)."*

Also, I did not find strong statement about why Cu was chosen in particular compared to other elements?

*Cu was chosen as a model of contaminant and choose for its wide used in agricultural practices as mentioned lines 52-53 "In particular, Cu is widely used as a fungicide, especially against downy mildew in vineyards (Komárek et al., 2010), but also in industrial processes"*

Another major limitation is related to the resolution of grid cells, i.e. 50 km (and what it means in terms of representation of runoff processes for instance). What are the limitations of using such a resolution and with calculations of mean runoff rates (in mm/yr) compared to actual processes affecting soils?

*You are right the grid cells used here were quite large. However, this is a common scale for land surface models that allows us to compare 2x2 couple of models and increase the confidence in our estimations. Moreover, due to the difficulty in climate prediction especially at fine scale, the use of finer scale models even closer to soil processes would have added another source of uncertainty. This has been precise lines 106-108:" The rainfall predictions were analyzed at the 0.5° that is a common scale for climate models allowing a multi-comparison to capture the variability in soil properties and rainfall regime"*

Overall, I think that the title/abstract/introduction/objectives should be rewritten to reflect better the actual content of the manuscript and these limitations (and discuss them explicitly).

*As mentioned above, title and introduction have been rephrased. (see first question). The need for intra annual assessment has been precise in the abstract lines 39-40 :" highlighting the global risk of combined climate change and contamination and the need for more local and seasonal assessment. Results are discussed to highlight the points requiring improvement to refine predictions." The discussion has also been improved lines 402-403 :" we chose to focus on the partitioning coefficient which is calculated based on soil properties (pH and OM here) others than total soil Cu."; lines 436-438 "Highly erosive storm events*

*predicted to increase during the next decades in Europe are another risk factor for freshwater contamination even in AP areas, but are often very punctual and local" lines 414-417 "In particular, there is large uncertainty about the C stocks that may change as a result of climate change and the policies for increasing these C stocks (Bruni et al., 2022). Besides, organic fertilizers applied to increase C stocks can change both pH and soil Cu content leading to supplementary uncertainties (Laurent et al., 2020)." Lines 461-462 "a larger Cu amount than locally computed taking into account total Cu and Kf may be exported through runoff", lines 462-464 "Thus, local soil Cu budgets require the use of temporal model, which accounts for the regular inputs and outputs of Cu from vegetation and runoff that cannot be accounting with multiyear mean" and lines 454-457 "It has already been identified that during high loads events, much more Cu was transported in solution than during light events (Imfeld et al., 2020) but alternations of drying and rewetting events may also affect Cu partitioning between phases (Christensen and Christensen, 2003; Han et al., 2001)."*

**Other remarks**
 *All the grammar corrections have been done within the manuscript. The rephrasing is indicated below*

All manuscript
Overall, I found the LP and AP acronyms very little indicative of what they are standing for, which complicates the reading (LP = contaminant leaching; AP = potential accumulation areas).
 *The introduction of these terms has been rephrased as: ": i) estimate the areas the most likely to lose soil Cu within soil solution and waterflows in Europe, thereafter, named leaching potential areas [LP], for the beginning of the century and ii) predict their changes according to different climate change scenarios. Additionally, we aimed to estimate the areas the most likely to accumulate Cu thereafter named accumulation potential areas [AP]."*

Overall, I would often replace the term 'concerns' throughout the manuscript; as it does not seem to be used in an appropriate way
 *This has been done throughout the manuscript*

**Abstract**
L.14 'besides pedological driven partitioning' >> unclear, please rephrase.
 *This has been removed*

L.17 please better introduce the choice of Cu
 *This has been detailed as:" We focused on copper (Cu) widely used in agriculture under mineral form or associated to organic fertilizers, resulting in high spatial variations in deposited and incorporated amounts in soils as well than in European policies of application"*

L.18 among contaminants
 *This has been removed*
L.26 why the median?

*We choose to focus on the median rather than on the mean as the median is less sensitive to extreme data -as might occur when dealing with contaminated soils. This has been precise line 114 :" median that is less driven by extreme than mean"*

LL.35-36 these scenarios are not introduced at all in the abstract
*The short name has been added line 175:" we performed our study with two representative atmospheric greenhouse gases concentration pathways (RCP)"*

**Introduction**

L.51 vineyard parcels
*Done*

L.63 vary not varied
*Done*

L.66 why Cu in particular?
*Some details have been added lines 53-54: "Besides, Cu application to soils are numerous, in the mineral form or within the organic fertilizers applied, leading to a global European limit of application"*

L.69 remove 'the' before 'decades'
*Done*

LL.71-72 unclear, please rephrase
*This has been rephrased as: "with extent of rain- and snow-fall alterations depending on climate change"*

L.73 the conversion between rainfall and water flows is not straightforward here
*We precised the soil waterflows lines 79-80." Thus, climate change will alter the soil waterflows throughout the century (Mimikou et al., 2000). "*

LL.76-78 this depends on the scale of interest…
*This has been rephrased "However, predicting how these runoff changes will relate to elemental contaminant fluxes in the coming decades remains difficult."*

LL.85-87 I may agree for current trends, but what about future trends?

L.88 Cu mobilility potential, but what about future Cu inputs?

*For the comments above, we do not understand which parts of the sentences are concerned. Are the line numbers correct?*

L.96 still, I would argue that the map with total Cu concentrations should be taken into account or at least compared to the model outputs (so that it can be discussed in terms of real risk not just potential behaviour?)

*The comparison with total Cu concentrations for the 21st century would require Cu inputs for which there is no reliable information in terms of estimations*

L.103 why the median?
*We precised it and the phrase now reads lines 113-114 :" by comparison between the local values of Kf and of runoff to the respective calculated European median that is less driven by extreme than mean"*

**Materials and methods**
L.122 how would you define the 'European level'?
*We specified "european Union level" that corresponds to the data provided by the joint research center*

L.126 each soil parameter values?
*This has been rephrased : "Have been fitted on a large range of each soil parameter"*

L.127 'in situ long term contamination' > do you mean 'monitoring' instead of 'contamination'?
*No, this means that we do not consider experimental spikes of contamination*

L.142 I wonder what is the meaning of representing runoff at such a scale?
*As mentioned above this allows us to compare the outputs of different models and to reduce the uncertainty inherent to the use of a single (finer scale) mode*
*l*
LL.159-60 'beginning', 'middle', 'end' >> not sure these are the best terms?
*These terms have been removed*

**Results**
L.215 'chilies soils' >> unclear what this means?
*This has been rephrased as: "arid soils from Chile"*

**Discussion**

LL.144-45 if there is a map available, what not using it in the current study?
*We did not use the Cu in solution map in this study because the values depend on soil total Cu and therefore on future inputs that are unknown.*

P.24 I found all this very speculative
*We are not sure to fully understand which part of this page was very speculative. But we modified the section 4.1 to make it more concrete, see for instance lines 414-418 ". In particular, there is large uncertainty about the C stocks that may change as a result of climate change and the policies for increasing these C stocks (Bruni et al., 2022). Besides, organic fertilizers applied to increase C stocks can change both pH and soil Cu content leading to supplementary uncertainties (Laurent et al., 2020)."*

L.176 '5 yrs-means' : the associated limitation should be further discussed (in terms of runoff driven by short intense events, this is a major limitation…)

*The paragraph has been rephrased lines 453-466 "It must be noted that the scope of our predictions had limits that rely on the difficulties to predict whether rain- and snow-falls and runoff will evolve in terms of intensity and frequency. It has already been identified that during high loads events, much more Cu was transported in solution than during light events (Imfeld et al., 2020) but alternations of drying and rewetting events may also affect Cu partitioning between phases (Christensen and Christensen, 2003; Han et al., 2001). Also, to gain field reality at the local scale such as landscape or catchment for example, modelling will require to account for the time periods of year with higher rain- and snowfalls amounts coinciding with periods of Cu use, for instance in agriculture and vineyards (Ribolzi et al., 2002; Banas et al., 2010). Indeed, if intense rainfall occurs close to Cu fungicide applications, a larger Cu amount than locally computed taking into account total Cu and Kf may be exported through runoff (Ma et al., 2006b, a). Thus, local soil Cu budgets require the use of temporal model, which accounts for the regular inputs and outputs of Cu from vegetation and runoff that cannot be accounting with multiyear mean. Finally, the identification of the areas with high risks of soil Cu leaching or accumulation we made in this study can be viewed as a first step for the risk change assessment of Cu contamination useful for land management or Cu-fertilizer applications regulations."*

---

## Author Response (AR2)

We thank the referee for his evaluation and comments. The last comments were considered carefully. Below are your comments in roman and our responses in italic font.

L33-34: our results simulate a constant global surface". Please rephrase (results do not simulate something and please explain what is meant by a constant global surface);

*We rephrased and write:" Interestingly, we simulate a constant surface area with"*

l. 67-68: "know and predict" = "knowing and predicting"

*OK*

l. 99-100: "at the 0.5° that" = "at a 0.5° resolution, which"

*OK*

l. 107: "driven by extreme than mean" = "affected by extreme values than the arithmetic mean" l. 135: "this research"= "this search"

*OK*

l. 151: "rain- and snow- falls" = "precipitation"

*OK*

l. 236: "JR'"" : please explain (Joint Research Centre?, if yes, then please add reference)

*Yes, this is JRC. The missing letter and the reference have been added*

l. 236: "sSauvé" = "Sauvé"

*OK*

l. 380: "which is calculated" = "which was calculated" l. 480: "t he"= "the"

*OK*